# Optimised Electroporation for Loading of Extracellular Vesicles with Doxorubicin

**DOI:** 10.3390/pharmaceutics14010038

**Published:** 2021-12-24

**Authors:** Angus J. Lennaárd, Doste R. Mamand, Rim Jawad Wiklander, Samir EL Andaloussi, Oscar P. B. Wiklander

**Affiliations:** 1Division of Biomolecular and Cellular Medicine (BCM), Department of Laboratory Medicine, Karolinska Institutet, Karolinska University Hospital Huddinge, 14157 Huddinge, Sweden; angus.lennaard@ki.se (A.J.L.); doste.mamand@ki.se (D.R.M.); rim.jawad@ki.se (R.J.W.); Samir.el-andaloussi@ki.se (S.E.A.); 2Department of Biomedical Sciences, Faculty of Science, Cihan University-Erbil, Kurdistan Region, Erbil 44001, Iraq

**Keywords:** extracellular vesicles, exosomes, doxorubicin, optimisation, electroporation, loading, chemotherapeutics, cytotoxins, intercellular transport, drug delivery platform, EV thawing temperature

## Abstract

The clinical use of chemotherapeutics is limited by several factors, including low cellular uptake, short circulation time, and severe adverse effects. Extracellular vesicles (EVs) have been suggested as a drug delivery platform with the potential to overcome these limitations. EVs are cell-derived, lipid bilayer nanoparticles, important for intercellular communication. They can transport bioactive cargo throughout the body, surmount biological barriers, and target a variety of tissues. Several small molecule drugs have been successfully incorporated into the lumen of EVs, permitting efficient transport to tumour tissue, increasing therapeutic potency, and reducing adverse effects. However, the cargo loading is often inadequate and refined methods are a prerequisite for successful utilisation of the platform. By systematically evaluating the effect of altered loading parameters for electroporation, such as total number of EVs, drug to EV ratio, buffers, pulse capacitance, and field strength, we were able to distinguish tendencies and correlations. This allowed us to design an optimised electroporation protocol for loading EVs with the chemotherapeutic drug doxorubicin. The loading technique demonstrated improved cargo loading and EV recovery, as well as drug potency, with a 190-fold increased response compared to naked doxorubicin.

## 1. Introduction

Extracellular vesicles (EVs) are lipid bilayer nanoparticles that can mediate cell-to-cell communication by the transport of membrane receptors and bioactive cargo, such as proteins, lipids, nucleic acids, and signalling molecules [1,2,3]. EVs are secreted under physiological conditions but can also be released in response to cellular activation, stress, or changes in the microenvironment, and both cell origin and the triggers for release, have been suggested to determine the content and the function of the EVs. [1,4,5,6]. They have been shown to cross biological barriers, such as the blood-brain barrier, and due to their great structural and compositional heterogeneity, implement a variety of functions in the recipient cell [7,8,9]. The cell specificity, mechanism of uptake, and intracellular release appear to be regulated by a range of molecular interactions between the EVs and its target cell [10,11,12,13].

EVs can distribute through the circulatory system and have been shown to extravasate through leaky vessels leading to accumulation in tissues with elevated blood supply and hyperpermeable vasculature [13,14,15,16]. These attributes promote EVs to naturally accumulate in tumour tissue, rendering it a suitable drug delivery platform for chemotherapeutics, where effector molecules can be loaded into the EVs and elicit a potent response in the cells of interest. Numerous small molecule drugs have been successfully incorporated as EV cargo and have shown to improve potency, increase accumulation in target cells, enhance drug stability and circulation time, resulting in a decreased IC_50_ [17].

Doxorubicin is a chemotherapeutic commonly applied to treat a variety of cancers including breast cancer, sarcoma, and leukaemia [18,19]. The drug is known to accumulate in cardiovascular tissue and is associated with several severe adverse effects such as fatigue, nausea, and an eight-fold increased risk of fatal cardiotoxicity [20,21]. Its cumulative dose is a critical parameter determining the cardiovascular effect, limiting both higher doses and prolonged treatments [22,23]. The use of EVs as carriers, allows for an altered biodistribution, shifting the ratio of accumulation from the cardiovascular tissue to tumour tissue, yielding a more efficient delivery allowing for lower dosages and subsequently reduced adverse effects [21,24]. Doxorubicin is bright red and exhibits a strong excitation signal, allowing for easy detection using fluorescent-sensitive instruments [25].

There are several endogenous and exogenous techniques applicable for loading EVs. The different methods have their own benefits and deficits, and the optimal loading technique varies upon the cargo and the experimental setting. Well-established methods include sonication, electroporation, extrusion, and freeze-thawing. However, all common methods are limited by an insufficient loading efficiency, restricting their applicability both in research and for clinical use. For loading therapeutic EVs with hydrophilic small molecule cytotoxins, such as doxorubicin, electroporation is commonly applied. It is often considered an advantageous method as it is easy to master, can be performed without toxic additives, and yields a fair loading efficiency relative to alternative established methods [17,21,26,27,28,29,30,31]. Electroporation is achieved by applying an external transmembrane electric field, which superimposed on the resting transmembrane potential, is greater than the dielectric strength of the membrane. With sufficient field strength under the right conditions, transient breakage in the bilipid layer can be induced, rendering a state of ephemeral membrane permeabilisation. This results in increased leakage of EV cargo but also increased uptake of surrounding molecules. By incubating the vesicles with the cargo of interest, the intended cargo may enter the lumen during the electric pulse, where they will be trapped upon membrane resealing [32,33]. The extent of membrane permeabilisation during electroporation can be regulated by the electrical pulse characteristics, dictating both loading efficiency and recovery of EVs [32,34].

As a smaller radius is correlated with increased membrane stability, it is recommended to apply higher field strength to achieve good membrane permeabilisation when working with smaller vesicles [35]. Stronger pulses might however induce damage on both the EVs and their cargo, and electroporation is associated with aggregation of nucleic acid species and loss of EV membrane integrity [26,28,36]. Nevertheless, it has been shown that adjustment of the electroporation conditions, can diminish these unfavourable effects [30]. Considering the frequent use of electroporation in the field, and the progress of EVs as nanocarriers, an optimised loading procedure, yielding higher loading efficiencies without inducing damage to either cargo or EV, is of great importance for future EV research.

This study aimed to improve current electroporation techniques for incorporating the chemotherapeutic drug doxorubicin into EVs, improving loading efficiency and increasing recovery without negatively affecting the innate qualities of EVs. We hypothesised that by systematically evaluating important parameters for electroporation, we could distinguish correlations and refined conditions, allowing us to design an improved electroporation protocol (Figure 1). The original loading procedure used in this study, along with the different adjustments, was based upon established protocols and previous research, and successively optimised by our findings [21,31,35,37,38,39,40,41,42,43,44,45]. The final product of doxorubicin-loaded EVs was further evaluated and compared to the naked drug as well as the commercially available pegylated liposomal doxorubicin, lipodox. The EVs in this study were isolated by size dependant methods targeting particles of 50–200 nm, and the samples thus consist of a compilation of small EV subpopulations within this size range.

## 2. Methods

### 2.1. Cell Culture

FreeStyle^TM^ HEK293F cells were cultured in serum-free FreeStyle^TM^ 293 Expression medium (Gibco Life Technology, Foster City, CA, USA) supplemented with 1% Antibiotic Antimycotic (Gibco Life Technology) and 4.5 g/L pyruvate (Invitrogen, Waltham, MA, USA). B16F10 cells were cultured in DMEM high glucose medium (DMEM; Gibco Life Technology, MA, USA), supplemented with 1% Antibiotic Antimycotic (Invitrogen) and 10% FBS (Gibco, Waltham, MA, USA). Mycoplasma testing was routinely performed, and all cell lines were maintained at 37 °C with 5% CO_2_ in a humidified atmosphere.

### 2.2. Isolating Small Extracellular Vesicles

EVs were isolated as previously described in [46]. In brief, once a concentration of 2 × 10^6^ ± 2 × 10^5^ 293F cells/mL was obtained, the condition media (CM) was collected and centrifuged at 700 RCF for five minutes to remove cells. The pellet was discarded, and the CM centrifuged for an additional 10 min at 2000 RCF to remove cell debris and additional larger contaminants. Subsequent purification by 0.22 µm filtration was performed (Nalgene Rapid-Flow; Thermo Fisher Scientific, Waltham, MA, USA) and the EVs were concentrated by tangential flow filtration (KrosFlo Research 2i TFF system; Spectrum labs) at a flow rate of 100 mL/min and a transmembrane pressure around 3 psi using a 300 kDa polyethersulfone hollow fibre filter (MidiKros, 370 cm^2^ surface area, SpectrumLabs, Hudson, MA, USA). The sample was diafiltered with twice the original CM volume using 0.22 µm filtered (Nalgene Rapid-Flow; Thermo Fisher Scientific) 0.1 M PBS and collected at a final volume of 20–35 mL. The concentrate was purified by 0.22 µm suction filtration (Nalgene Rapid-Flow; Thermo Fisher Scientific) and the filtrate was further concentrated by ultrafiltration at 4000 RCF in 10 kDa spin columns (Amicon Ultra-15; Millipore, Burlington, MA, USA) to a final volume of 400–600 µL.

### 2.3. Quantification of Extracellular Vesicles

The Brownian motion of the particles within the size range of 10–2000 nm was analysed by a nanoparticle analyser (NanoSight NS500; Malvern Panalytical, Malvern, UK), and the nanoparticle size distribution and concentration were calculated using software (NanoSight NTA v. 2.3; Malvern Panalytical). To avoid particle track intersection, and allow for differentiation of individual particles during analysis, the samples were diluted in 0.22 µm filtered (Nalgene Rapid-Flow; Thermo Fisher Scientific) 0.1 M PBS to a maximum concentration of 2 × 10^9^ particles/mL. A concentration of a minimum of 2 × 10^8^ particles/mL was used to ensure an adequate number of particles for sufficient statistical power. To provide comparable readings with strong signal and low background noise, the camera level was kept at 14 for all samples, and the camera focus was set manually to capture images with each particle visible as sharp dots. The point scatter was recorded in five videos of 30 s, divided by advancing the sample three times, and a delay of 5 s. The analysis was performed with the screen gate set at 10 and the detection threshold at 7 and with all remaining settings set at automatic.

### 2.4. Protein Detection by Western Blot

The presence of the EV-associated tetraspanin CD81 and intravesical protein Alix was examined by Western blot. 2 × 10^6^ 293F cells from the production culture were washed with PBS and pelleted at 300 RCF for 5 min. The cell pellet and 2 × 10^9^ EVs were lysed separately using 100 µL of radioimmunoprecipitation buffer (RIPA; BioRad, Hercules, CA, USA). Both samples were incubated on ice for 30 min and vortexed for 10 s each 5th minute. Lipids and additional large contaminants were removed from the cell lysis by centrifugation at 12,000 RCF for 12 min at 4 °C. 24 µL of supernatant was transferred to a new tube on ice. The cell lysis was centrifuged at 12,000 RCF for 12 min at 4 °C. 24 µL of the supernatant and 24 µL of EVs were mixed with 8 µL loading buffer (10% glycerol, 8% Sodium dodecyl sulphate, 0.5 M dithiothreitol, and 0.4 M sodium carbonate), incubated at 65 °C for 5 min prior to being loaded onto the NuPAGE+ (Invitrogen, Novex 412% Bis-Tris gel) and run at 120 V for 2 h. The proteins were transferred by iBlot system (iBlot 2 Dry Blotting System; Invitrogen) for 7 min to an iBlot membrane (iBlot 2 Transfer Stacks; Invitrogen). The membrane was treated by blocking buffer (Odyssey Blocking Buffer; LI-COR Biosciences, Lincoln, NE, USA) at room temperature for 60 min and subsequently incubated overnight at 4 °C with newly prepared primary antibodies (anti-CD81 (A1816) diluted 1:1000, anti-Alix (K1115) diluted 1:1000). The membranes were washed four times with TBS-T 0.1% for 5 min each on a shaker and then incubated with secondary antibody (Goat anti-Mouse (C00322) diluted 1:10,000, Goat anti-Rabbit (C90827-25) diluted 1:10,000) in RT for 1 h. The washing was repeated with PBS.

### 2.5. Loading Extracellular Vesicles

The number of EVs, the concentration of doxorubicin hydrochloride (Sigma Aldrich; Merck, Germany), the composition of electroporation buffer as well as characteristics of the electric pulse were altered between the experiments and are described in detail in the result section, only the common practices will be explained further here.

Doxorubicin was diluted to a concentration of 10 mM with ultrapure water, to avoid aggregation at later stages. The doxorubicin was mixed with EVs diluted with 0.22 µm filtered (Nalgene Rapid-Flow; Thermo Fisher Scientific) 0.1 M PBS and incubated at 4 °C for 30 min. Prepared electroporation buffer was added to the EVs in a 1:1 ratio, and 400 µL of the sample was electroporated in 0.4 cm cuvettes by exponential pulse using an electroporation system (GenePulser Xcell; BioRad). The sample was then incubated at 37 °C for 30 min, and the EVs were subsequently isolated by size exclusion chromatography (SEC), using 70 nm 500 µL qEV columns (qEVoriginal; IZON Science LTD, Christchurch, New Zealand) collecting the 4th and 5th ml of elute. The elute was then concentrated to a final volume of 200–300 µL, using 10 kDa spin filters (Amicon Ultra; Millipore) spun at 4000 RCF.

### 2.6. Electroporation of Naked Doxorubicin

The loading procedure for the control of naked doxorubicin was performed in absence of EVs and without the isolation or ultrafiltration steps. The voltage, capacitance, and electroporation buffer were altered between the procedures, further described in the result section. A sample from the original mixture of doxorubicin and electroporation buffer was left without further processing and used as a control validating the intended original doxorubicin concentration.

### 2.7. Quantification of Doxorubicin

Doxorubicin was analysed using a fluorometer (SpectraMax^®^ i3x; Molecular Devices LLC, San Jose, CA, USA) and the concentrations were determined in relation to a standard curve starting at 100 µM halving 12 times to 40 mM and blanks of ultrapure water and PBS, using the dedicated software (SoftMax Pro v.7; Molecular Devices LLC). The samples were portioned at 90 µL and excited at 488 nm and read at 530 nm. Effective loading was calculated as a function of mM doxorubicin per billion EV.

### 2.8. Viability Assay

1 × 10^4^ B16F10 cells were seeded in a 96 well plate 24 h prior to treatment, and the viability was assessed 48h post-treatment by Cell Titre Glo (CellTiterGlo; Promega Biotech, Madison, WI, USA) following the provided protocol on a luminometer (GloMax 96 Microplate Luminometer: Promega Biotech). The plate was read each third minute until the signal had stabilised. The lipodox (Caelyx; Schering-Plough, Kenilworth, NJ, USA) dose was calculated based on the concentration provided by the manufacturer.

### 2.9. Data Analysis

The data were assumed parametric and analysed using R (R Project v.4.1; R Core Team, Vienna, Austria) with α at standard 0.05. Statistical significance is illustrated as follows: * *p* < 0.05; ** *p* < 0.01; *** *p* < 0.001, **** *p <* 0.0001, and the graphs composed by Prism (Prism v.8.4.2; GraphPad Software, San Diego, CA, USA). Significance is indicated where the findings are significant and of importance for the study. If no statistical values are presented, the findings should be assumed non-significant.

## 3. Results

### 3.1. Preparatory Analysis of the Methods Applied 

The isolation of EVs was verified by nanoparticle tracking analysis (NTA) and Western blot (WB). The NTA validated the presence of nanosized particles within the mode size range of 115 ± 10 nm (Figure 2A), and WB confirmed enrichment of EV markers CD81 and Alix (Figure 2B).

To determine the optimal thawing temperature for EVs stored at −80 °C, samples were thawed and refrozen in triplicates through three cycles at room temperature (RT), 4 °C, and on ice. The results indicate that up to twice the amount of EVs could be recovered when thawed at RT as compared to on ice (*p* < 0.001) (Appendix A).

A multistep purification process consisting of size exclusion chromatography and spin filtration was applied to remove unloaded doxorubicin and concentrate the sample post-electroporation. Prior to optimisation, it was validated that the method of purification efficiently removed naked doxorubicin from the sample, allowing for accurate quantification of EV encapsulated doxorubicin. Five controls with 3 mM doxorubicin were treated following the electroporation process described in an established protocol, using a buffer of 600 mM sucrose and a pulse setting of 250 V and 125 µF (19). The sample was subsequently mixed with EVs, and the purification process was then applied. The product was analysed by fluorometry demonstrating the removal of doxorubicin in the collected samples, yielding a negative effective loading of −0.16 µM/1 × 10^9^ EVs.

The cargo loading was quantified by fluorometry, and the effective loading was calculated as a function of µM of doxorubicin per EVs in a given volume, determined by NTA. To ensure that the method of quantifying cargo loading did not give a false positive result mediated by the EVs per se, an additional control with 5 × 10^11^ EVs without doxorubicin was prepared, and electroporated at 250 V with 125 µF and analysed by NTA and fluorometry. The recovery was noticeably lower in the control samples compared to the samples with doxorubicin, and the total loading was determined to be below the detection threshold (0.08 µM, yielding a theoretical effective loading of 0.003 µM/1 × 10^9^ EVs).

### 3.2. Optimisation of Parameters Important for Successful Electroporation

Important parameters for efficient electroporation include sample composition. To maximise material efficiency for later steps, the relative concentrations of EVs and doxorubicin were primarily optimised. To do this, electroporation was carried out using 5 × 10^11^, 1 × 10^11^, or 5 × 10^10^ EVs with 2.0 mM of doxorubicin in the conditions described in the established protocol, using an electroporation buffer containing 600 mM sucrose in PBS and a pulse setting of 250 V and 125 µF [21]. The recovered EVs were quantified using NTA and the doxorubicin concentration was determined by fluorometry. Both the proportion of recovered EVs and doxorubicin in the samples were significantly elevated (*p* = 0.03, *p* < 0.001) in the higher EVs concentrations (5 × 10^11^) compared to the lower concentrations (5 × 10^10^) (Figure 3A,B). However, the effective loading per EV indicated a peak in samples of 1 × 10^11^ EVs with 0.42 µM/1 × 10^9^ EV (Figure 3C). This was more than twice the efficiency compared to the second and third conditions; 5 × 10^10^ at 0.18 µM/1 × 10^9^ EVs and 5 × 10^11^ at 0.17µM/1 × 10^9^ EV.

To further evaluate the relationship between the concentrations of cargo and EVs, 5 × 10^11^ EVs were electroporated with 3 mM, 2 mM, 1 mM, 0.5 mM, and 0.25 mM of doxorubicin. Both the total loading and effective loading decreased significantly (*p* < 0.001, *p* = 0.03) at concentrations lower than 1 mM, while it remained unaltered above 1 mM (Figure 3E,F). There was no significant variability of recovery (*p* = 0.87), but the greatest effective loading was seen at 1 mM doxorubicin per 5 × 10^11^ (Figure 3D,F).

In addition, another control was added to confirm that the electroporation step was essential for the procedure to increase the loading beyond that of incubation alone. The control was identical to the sample of the highest concentration and was treated alongside the remaining samples excluding the electroporation. The electroporated samples had a higher loading compared to the control, however, no major differences in recovery could be observed (Figure 3A,B,D,E). This control was further repeated for all electroporation experiments and is further referred to as “CONTROL” in the Figures. 

Next, eight different electroporation buffers were evaluated regarding recovery and loading efficiency (Table 1). Five buffers were chosen from literature and previous research [21,34,36,37,38,39] while buffers 1, 2, and 4 analysed the impact of sucrose concentration, compared to buffer 3 (Table 1). All samples were electroporated with 1 × 10^11^ EVs and 1 mM doxorubicin per 5 × 10^11^ EVs at 250 V with 125 µF.

Electroporation buffer 8 had a noticeably superior recovery and total loading compared to the other buffers, yielding a 20% higher effective loading compared to the original electroporation buffer 3, used in previous steps. The highest effective loading was observed with buffer 2 with a 50% improvement (Figure 4B,C), and a distinct negative correlation between sucrose concentration and recovery could be observed for buffers 1–3 and with a minor improvement for buffer 4 (Figure 4A).

Samples electroporated with trehalose-based buffers 5 and 6, had a recovery of 154% and 119% compared to the original buffer (B3), respectively (Figure 3A). However, the effective loading was only 75% for both buffers, when compared to the original buffer (Figure 4C). Electroporation buffer 7 had 145% improved recovery compared to the original buffer (B3), but an inferior effective loading of 46% as compared to the original buffer (Figure 4A,C).

Succeeding optimization of sample composition, a series of electroporation pulse settings were evaluated in relation to loading efficiency. The extent of membrane permeabilization is dependent upon the electric charge stored in the system, measured as capacitance, and the field strength. As the field strength is a function of voltage and the distance between the electrodes, keeping the position of the electrodes with a constant distance of 4 mm, the field strength could be altered by changing the voltage. Accordingly, the effect of alternative pulse characteristics was analysed using a total of 9 sequences of voltage and capacitance (Table 2) [31,40,41,42,43,44], using 1 × 10^11^ EVs and 1 mM doxorubicin per 5 × 10^11^ EVs, in the two superior electroporation buffers B2 and B8, respectively.

Noticeably, there were individual settings that outperformed, yielding improved recovery and increased loading (Figure 4). Electroporating in buffer 2 at a pulse setting of 950 V:50 µF yielded superior total loading, and a pulse setting of 125 V:900 µF delivered the greatest effective loading while still maintaining a decent recovery. Buffer 8 had superior total and effective loading at a pulse setting of 950 V:50 µF and 250 V:125 µF. All three settings gave adequate results in their respective sample in its alternative buffer. Furthermore, a significant negative correlation (*p* < 0.001) between recovery and effective loading (Figure 5A,B) could be observed, where reduced recovery was associated with improved effective loading (Figure 5D).

It was noted that samples electroporated in buffer 2 exhibited a colour change from red to purple, at 250 V:500 µF, 250 V:950 µF, 500 V:500 µF, and 950 V:50 µF. The same could be observed for samples electroporated with buffer 8 at 250 V:950 µF and 500 V:500 µF. Moreover, the samples electroporated in buffer 8 at 500 V:950 µF caused an arc discharge, resulting in a loss of 99% of the EVs. Alterations in how compounds absorb and reflect light are dependent upon changes in their chemical properties and could consequently affect bioactivity [47]. The effect was therefore considered undesirable, and all affected conditions were discontinued from further optimisation.

### 3.3. Evaluation of the Optimised Loading Conditions

To validate that the new electroporation protocol mediated functional EVs equal or better than that of the original protocol, an in vitro viability assay was performed using 2.5 × 10^11^ EVs with 1 mM doxorubicin per 5 × 10^11^ EVs in buffer 2 and 8 at 125 V:900 µF, 250 V:125 µF, and 950 V:50 µF and compared to the original protocol using buffer 3 at 250 V:125 µF, represented as mode 1 (Table 3). Cells were treated in triplicate using 1 × 10^10^, 1 × 10^9^, and 1 × 10^8^ EVs. The relative cell viability was analysed using Cell Titre Glo that measures cellular production of ATP.

The recovery of EVs when loading by mode 7, yielded a recovery of 9.73%, leading to an insufficient concentration for the viability assay dosed at 1 × 10^10^ (Figure 6A). This condition was known, given our previous data, to yield the lowest recovery of the modes evaluated, but was included for further evaluation owing to its promising loading profile (Figure 6B). The samples prepared with mode 7 were still added at lower concentrations, 1 × 10^9^ and 1 × 10^8^.

The relative effect, counted as the relative cell viability per µM doxorubicin added to the well, was calculated for all treatments. The greatest relative effect was achieved with mode 2, yielding doxorubicin-loaded EVs with a 190-fold increased effect compared to naked doxorubicin (*p* < 0.001), twice the relative effect compared to lipodox (*p* = 0.019), and a 20% improvement compared to the originator loading protocol, mode 1 (Figure 6D).

It was noted that EVs loaded by mode 6 exhibited an altered potency profile as analysed by the viability assay compared to the remaining samples. The samples from mode 6 exhibited an inferior effect compared to the other samples in all concentrations, with 10% viability at 1 × 10^10^ EVs, 46% at 1 × 10^9^ EVs, and 84% at 1 × 10^8^ EVs. In comparison, the relative cell viability ranged between 1–3% at 1 × 10^10^ EVs for all other doxorubicin-loaded EV samples. The variance for the remaining modes were 4–13% at 1 × 10^9^ and 14–22% at 1 × 10^8^ EVs (Figure 6C).

The loading conditions were then assessed by further characterisation of the colour change, evaluating the impact of which the loading procedure might inflict upon doxorubicin. The doxorubicin was electroporated in conditions similar to the loading procedure, using three different modes. One known to alter the colour of the sample from red to purple (Mode 9), one following a previously established protocol (Mode 1) [21], and one following the optimised conditions (Mode 2). The electroporated product was analysed by fluorometry and the potency was evaluated in vitro.

Changes in a compound’s chemical structure, such as the destruction of covalent or non-covalent bonds, alter the compound’s fluorescent properties [48]. As the chemical structure is essential for the drug’s molecular interaction, and thus bioactivity, analysing the fluorescent alterations can illuminate unwanted changes to the cargo. By comparing the fluorescent profile of unprocessed doxorubicin to doxorubicin treated by the three different modes, we could demonstrate that the mode of loading had a distinct impact on the fluorescent properties of the sample. Mode 9 reduced the fluorescence on average 47%, mode 1 reduced the fluorescence on average 22%, and mode 2 reduced the fluorescent signal on average 2% (Appendix A).

These results indicate that the mode of loading can inflict molecular alteration to doxorubicin, and it is therefore of importance to further elucidate potential changes in the bioactivity of the drug. The potency was evaluated by Cell Titre Glo viability assay as previously described, where doxorubicin processed by the three different modes were applied in three different doses and compared to unprocessed doxorubicin, at the same concentrations determined by fluorometry (Appendix A). The relative cell viability in the well treated with the processed doxorubicin was then individually compared to the relative cell viability in the wells treated with unprocessed doxorubicin. A noticeable variation in potency could be observed for doxorubicin treated by mode 1, with a 1.70-fold deviation compared to the unprocessed doxorubicin (Appendix A). The deviation was less pronounced when treated according to mode 9, with a 0.57-fold deviation, and mode 2 with a 0.38-fold deviation. This indicated that loading could affect the potency of the drug, even if the effect was not necessarily related to the observed colour change. The colour change would however affect the fluorescent readout, thus reducing the measured recovery.

Different loading procedures affect the samples in different ways, and it is therefore of utmost importance to ensure that the optimisation has sifted for improved loading, and not for a procedure with a decreased clearing of naked doxorubicin, increasing the fluorescent readout. To validate that the increased recovery of doxorubicin in the optimised loading procedure was due to improved loading of the EVs, and not a result of insufficient purification caused by the altered sample compositions, an additional control was added. Doxorubicin and EVs were electroporated separately in accordance with the optimised procedure, mode 2, and subsequently mixed before being purified by the multistep filtration method described in the methods section. As expected, fluorescent analysis of the samples demonstrated a 100% decrease of the fluorescent signal, demonstrating that the improved loading was not due to decreased removal of naked doxorubicin (Appendix A).

The versatility of the optimised loading procedure was further assessed by evaluating the applicability of the protocol on EVs derived from the additional cell source B16F10. 48.0% of the EVs were recovered after the loading process and 0.93 µM was successfully loaded, yielding an effective loading of 1.36 × 10^11^ nM doxorubicin per EV (Figure 7A,B). There was no significant difference from the previously used cell line, 293F, which had a recovery of 72.6%, and yielded a loading of 1.23 µM with an effective loading of 1.44 × 10^11^ nM per EV (Figure 7A,B).

The potency of the newly loaded B16F10 derived EVs was further assessed in vitro and compared to 293F derived EVs, naked doxorubicin, and lipodox. While there was no significant difference (*p* = 1.00) between the two types of EVs, both EV types presented a significant advantage compared to naked doxorubicin (*p* = 0.002) and nearly twice the cytotoxic effect of lipodox (*p* = 0.019) (Figure 7C).

## 4. Discussion

High drug encapsulation is a requirement for the effective use of EVs as nanocarriers, and for loading small molecule drugs, electroporation is commonly applied [27,29]. However, the efficiency of loading is often inadequate and improved procedures are required. The clinical importance, along with its fluorescent and chemical properties, renders doxorubicin a suitable drug for studying cargo loading of EVs. Here, we optimised important parameters for incorporating therapeutic doxorubicin into EVs, improving both cargo loading and EV recovery, as well as drug potency.

The preparatory analysis indicated successful EVs isolation, in line with previous findings following similar isolation techniques [46,49]. It was further demonstrated that the post-electroporation purification effectively removed naked doxorubicin and that the method of measurement accurately quantified both doxorubicin and EV concentration, which allowed us to properly evaluate loading efficiency.

Rapid thawing has been suggested to preserve the integrity of cellular membranes by minimising osmotic stress and ice recrystallisation [50]. Our data showed that rapid thawing in RT could increase EV recovery by up to 50% compared to thawing on ice, suggesting that this theory is valid for smaller lipid bilayer vesicles too. Furthermore, the positive correlation between the initial concentration of EVs, and both proportional recovery and total volume of loaded doxorubicin, clearly indicated that both EV and cargo input are crucial parameters influencing the success of loading. Our findings suggest that effective loading increases in tandem with an increased drug to EV ratio, up until 1 mM:5 × 10^11^ where it reaches a plateau. A pattern possibly owing to a limited loading capacity of EVs. Furthermore, it is worth noting that the ratio used for electroporation varies significantly within the field, where a similar protocol utilises a ratio of 12.5 mM:5 × 10^11^, highlighting the importance of optimisation. [21]. The greatest effective loading equals the most efficient use of material, it is however worth noting, that higher EV concentrations yield a greater proportional recovery and total loading. While the recovery of EVs was satisfactory when loading in trehalose-based buffers (B5, B6), suggested to stabilise EVs during and after electroporation [37,38], the loading was inadequate when compared to the alternative buffers. The most promising loading efficiency was instead observed using buffer 2 containing 400 mM sucrose, which further yielded a similar EV recovery. Electroporating 1 × 10^11^ EVs in a ratio of 1 mM:5 × 10^11^, with an electric pulse of 950 V and 50 µF in buffer B2 of 400 mM sucrose, the EV recovery was improved by 20% and the loading by 18%, compared to EVs loaded by the original protocol. It was further demonstrated that the EVs loaded using the optimised protocol maintained their capacity to carry and deliver therapeutic cargo in vitro, suggesting that the EVs preserved their integrity and desired biological function. The versatility of the optimised protocol in regard to EV cell origin was demonstrated by employing the method of loading on B16F10 derived EVs, yielding both recovery and effective loading similar to that of 293F derived EVs. Moreover, EVs derived from both cell lines performed similarly in vitro, further supporting the applicability of the protocol.

It was noted that certain electroporation conditions induced a change of colour in the samples, from red to purple. A similar change of colour has previously been noted for naked doxorubicin. The effect was then attributed to chemical decomposition, suggested as a result of cleaved amino sugars important for the effective moiety, thus altering the chemical properties and possibly bioactivity of the drug [51,52]. It can be hypothesised that this effect is caused by the electroporation cuvettes, as it has been shown that the electrodes release aluminium cations into the sample during intense electric pulses and that doxorubicin change colour upon excessive contact with aluminium [32,51,52,53]. However, as a similar effect of doxorubicin has been associated with a variety of stimuli, including extreme pH, light, and alkaline interactions, a range of alternative explanations remains possible [18,51,52]. By evaluating doxorubicin treated by different loading procedures we demonstrated that the condition of loading could affect both the fluorescent characteristics and efficacy of naked doxorubicin. Our results further showed that the optimised method had a negligible impact on the abovementioned properties. This highlights the importance of considering cargo degradation even when working with stable molecules, and further shows the advantages of our described optimised protocol.

Given the intrinsic cell targeting capability and inherent biocompatibility, EVs serve as a promising drug-delivering platform that can mediate effective transport targeting specific tissues and cells [6,11,12,54]. In this study, we systematically optimised an established electroporation protocol, improving loading efficiency, recovery, and drug potency in vitro, accomplishing a 190-fold increased drug potency compared to naked doxorubicin, and twice the potency of lipodox, a liposomal form of doxorubicin. Our results support the findings of previous studies demonstrating the superior therapeutic potency of small molecule chemotherapeutics when incorporated into EVs compared to the free drug in vitro and are of value for future research as well as it illustrates the platform as a promising treatment option for cancer patients in the future.

## Figures and Tables

**Figure 1 pharmaceutics-14-00038-f001:**
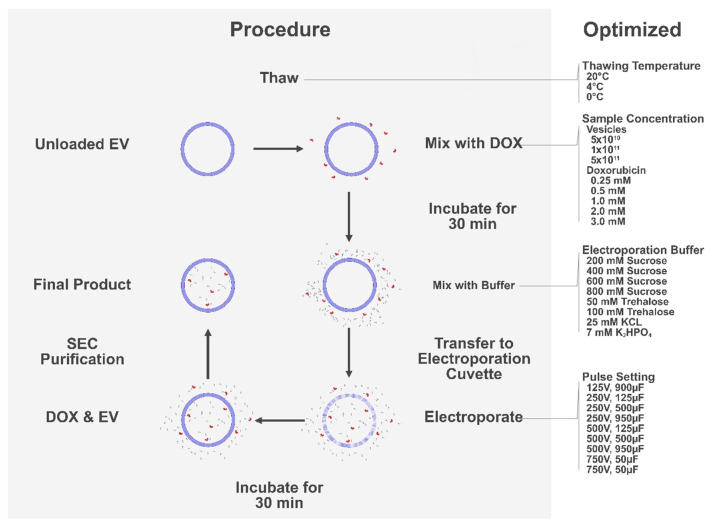
The process of optimisation. A schematic of the loading procedure including all steps of optimisation.

**Figure 2 pharmaceutics-14-00038-f002:**
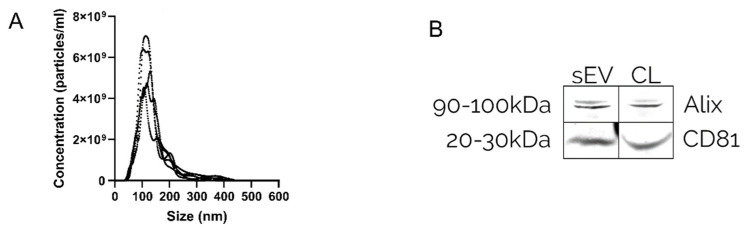
Characteristics of isolated EV sample. (**A**) A size distribution curve of an isolated EV batch determined by NTA, with the mode size of 109 ± 3.9nm, SD: 5.1 ± 2.0nm, and a concentration of 5.21 × 10^12^ particles/mL. (**B**) Western blot indicating the enrichment of EV markers CD81 and Alix in isolated sEV batch as well as cell lysate (CL).

**Figure 3 pharmaceutics-14-00038-f003:**
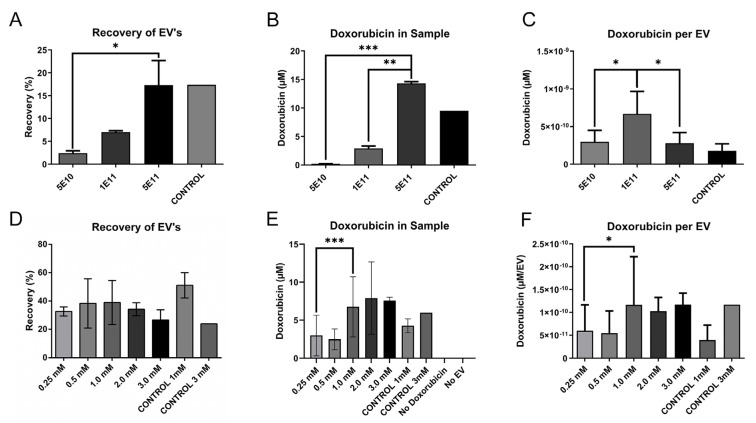
The concentration of EV and cargo has a significant impact on loading efficiency. (**A**) Recovery of EVs when loading with different numbers of EVs electroporated, with an un-electroporated control with 5E11 EVs. Measured by NTA (output/input). (**B**) µM doxorubicin in the final sample when loading with different numbers of EVs electroporated, with an un-electroporated control with 5E11 EVs. (**C**) Effective loading of EVs when loading with different numbers of EVs electroporated, with an un-electroporated control with 5E11 EVs. (**D**) Recovery of EVs when loading with different concentrations of doxorubicin, with an un-electroporated control with 3 mM doxorubicin. Measured by NTA (output/input). (**E**) µM doxorubicin in the final sample when loading with different concentrations of doxorubicin, with an un-electroporated control and a control with no doxorubicin. The “No Doxorubicin” and “No EV” controls are treated according to the loading procedure in the absence of respective components, doxorubicin and EVs. (**F**) Effective loading of EVs when loading with different concentrations of doxorubicin, with an un-electroporated control with 3 mM doxorubicin. Significance calculated by ANOVA and is illustrated as follows: * *p* < 0.05; ** *p* < 0.01; *** *p* < 0.001.

**Figure 4 pharmaceutics-14-00038-f004:**
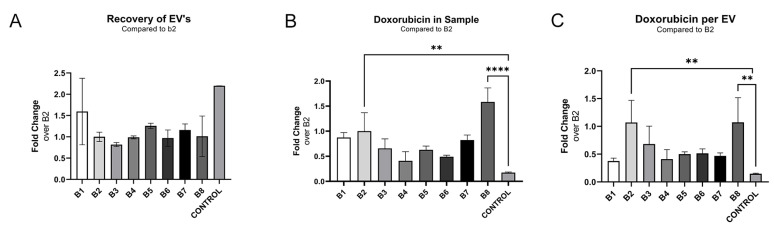
The composition of the electroporation buffer affects EV loading efficiency. (**A**) Recovery of EVs when loading with different electroporation buffers. Measured by NTA (output/input) and normalised towards buffer 2 of 400 mM sucrose. The buffers are noted as B1–B8 and are described in further detail in Table 1. (**B**) µM doxorubicin in the final sample, when loading with different electroporation buffers. (**C**) Doxorubicin per EVs normalised towards the buffer, B2 of 400 mM sucrose, when loading with different electroporation buffers. Significance calculated by ANOVA and is illustrated as follows: ** *p* < 0.01; **** *p* < 0.0001.

**Figure 5 pharmaceutics-14-00038-f005:**
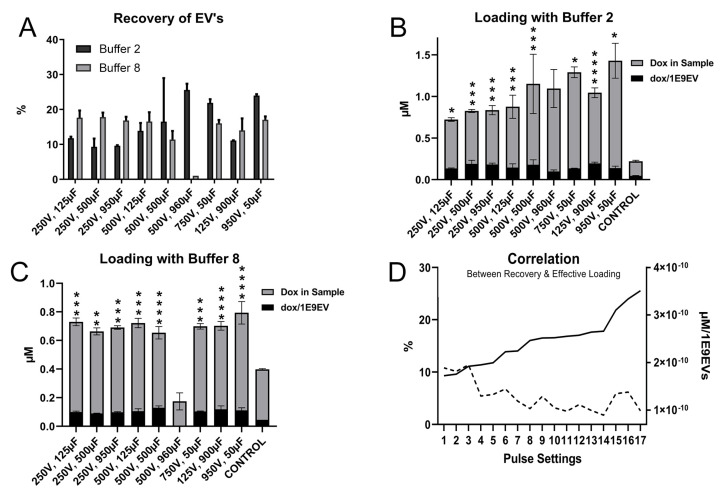
The pulse characteristics, regulated by applied voltage and capacitance, affects EV loading efficiency (**A**) Recovery of EVs when loading with different settings of voltage (V) and capacitance (µF). Measured by NTA (output/input). (**B**) Total doxorubicin in the final sample and µM doxorubicin per EVs b when loading with different settings of voltage (V) and capacitance (µF), electroporated with buffer 2 of 400 mM sucrose in PBS. Significance as compared to CONTROL. (**C**) Total doxorubicin in the final sample and µM doxorubicin per EVs when loading with different settings of voltage (V) and capacitance (µF), electroporated with buffer 8 of 272 mM sucrose, 7 mM di-Potassium hydrogen phosphate, and 1 mM magnesium chloride in PBS. Significance as compared to CONTROL. (**D**) A correlation graph demonstrating the significant relationship (*p* < 0.001) between recovery and effective loading, when electroporating with different electrical pulses regulated by voltage and capacitance. The right *y*-axis represents effective loading, corresponding to the dotted line. The left *y*-axis represents recovery, corresponding to the solid line. Significance calculated by ANOVA and is illustrated as follows: * *p* < 0.05; ** *p* < 0.01; *** *p* < 0.001; **** *p* < 0.0001.

**Figure 6 pharmaceutics-14-00038-f006:**
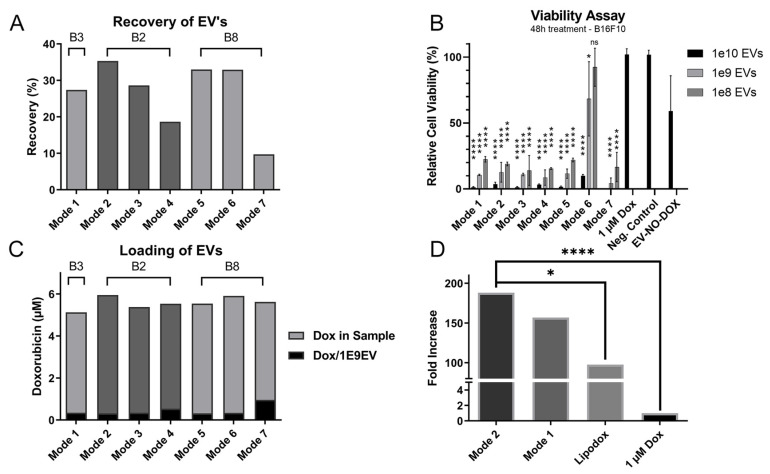
The optimised protocol improves the potency of doxorubicin-loaded EVs (**A**) Recovery of EVs when loading in different conditions, regulated by voltage (V), capacitance (µF), and electroporation buffer. Measured by NTA (output/input). The conditions are further described in Table 4, and the buffers are noted as B2, B3, and B8 and are described in further detail in Table 1. (**B**) Relative cell viability after 48h incubation with the different treatments. The significance is compared to 1 µM doxorubicin control. There was also a significant difference between all modes and EV-NO-DOX at 1E10 Evs, and for all modes except mode 6 at 1E9 and 1E8 EVs. (**C**) µM doxorubicin in the final sample and effective loading when loading using different conditions regulated by voltage (V), capacitance (µF), and electroporation buffer. (**D**) Fold increase of potency per µM of effector molecule doxorubicin between different treatments. Significance calculated by ANOVA and is illustrated as follows: ns, *p* > 0.05; * *p* < 0.05; **** *p* < 0.0001.

**Figure 7 pharmaceutics-14-00038-f007:**
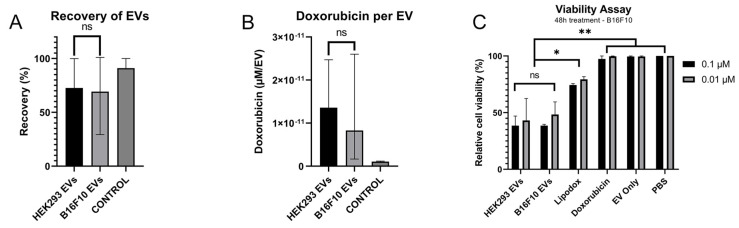
The protocol is applicable on additional EV cell source B16F10 and shows a superior cytotoxic effect in vitro compared to lipodox and doxorubicin. (**A**) Recovery of EVs, when loading different EV cell sources, with a control of B16F10 EVs. Measured by NTA (output/input). (**B**) Loading of EVs when loading different EV cell sources. (**C**) Relative cell viability after 48 h incubation with the different treatments. Significance calculated by ANOVA and is illustrated as follows: ns, *p* > 0.05; * *p* < 0.05; ** *p* < 0.01.

**Table 1 pharmaceutics-14-00038-t001:** Evaluation of the different electroporation buffers. List of the buffers evaluated.

Buffer	Composition
B1	Electroporation buffer of 200 mM sucrose in PBS.
B2	Electroporation buffer of 400 mM sucrose in PBS.
B3	Electroporation buffer of 600 mM sucrose in PBS [21].
B4	Electroporation buffer of 800 mM sucrose in PBS.
B5	Electroporation buffer of 50 mM trehalose in PBS [37,38,39].
B6	Electroporation buffer of 100 mM trehalose in PBS [38,39].
B7	Electroporation buffer of 1.15 mM potassium phosphate, 25 mM potassium chloride and 21% OptiPrep in PBS [40].
B8	Electroporation buffer of 272 mM sucrose, 7 mM di-Potassium hydrogen phosphate (adjusted to pH 7.4 with phosphoric acid) and 1 mM magnesium chloride in PBS [35].

**Table 2 pharmaceutics-14-00038-t002:** Evaluation of different pulse characteristics. Table of voltages (V) and capacitances (µF) evaluated.

Settings	50 µF	125 µF	500 µF	900 µF	950 µF
125 V				X	
250 V		X	X		X
500 V		X	X		X
750 V	X				
950 V	X				

**Table 3 pharmaceutics-14-00038-t003:** Evaluation of the impact of different loading procedures on the properties of doxorubicin. A table of the specific conditions used in the three procedures analysed.

Mode	Voltage	Capacitance	Buffer
1	250 V	125 µF	3
2	950 V	50 µF	2
3	125 V	900 µF	2
4	250 V	125 µF	2
5	250 V	125 µF	8
6	950 V	50 µF	8
7	125 V	900 µF	8

**Table 4 pharmaceutics-14-00038-t004:** Evaluation of the impact of different loading procedures on the potency of doxorubicin-loaded EVs in vitro. A table of the specific conditions used in the procedures analysed.

Mode	Voltage	Capacitance	Buffer
1	250 V	125 µF	3
2	950 V	50 µF	2
9	250 V	950 µF	2

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
