# Peer review of "Optimised Electroporation for Loading of Extracellular Vesicles with Doxorubicin"

_pharmaceutics, 2021, doi:10.3390/pharmaceutics14010038_

Round 1
Reviewer 1 Report
This paper entitled “Optimised electroporation for loading of extracellular vesicles with therapeutic cargo” describes a good preliminary study for optimising for the loading of extracellular vesicles with chemotherapeutic agent, doxorubicin. Whilst these findings have great potential and could provide insight and guidance for the therapeutic loading of EVs, there are a few concerns the authors should address.
Major concerns:
- Study rationale
- In the introduction, authors have neglected to cover the gap in which they set the rationale for the experiments within the paper. I highly encourage the authors include (1) an overview of electroporation as a drug loading technique, (2) the pros/cons of electroporation, (3) why electroporation is better to optimise in comparison to other drug loading techniques, and lastly (4) why does it need optimisation (i.e is the process not efficient enough?) as the reason why they’re pursuing electroporation is unclear.
- Data interpretation and experimental methodologies:
- Given the scope of the paper and questions addressed, it is as though the authors have over-interpreted their findings to be more generalised.
- Lack of consistent statistics (error bars, statistical tests) throughout the paper makes it difficult to reconcile the findings.This must be included in each figure legend, otherwise it affects credibility and clarity. For instance, how did the authors draw conclusions such as whether a particular electroporation programming is ‘best’ when there is no statistics shown?
- Additionally due to lack of consistent statistics, authors tend to over interpret their data. There are many instances of this: (1) Comparing the loading efficiency of a published concentration (12.5mM:5x1011) to theirs (1mM:5x1011) (Page 12, Line 24), which is strange when the previous drug concentration was not addressed through their experiments. (2) Lack of consistent and appropriate controls (figures 2, 5, and 6). Without a dox-spiked control, they cannot be certain that doxorubicin is being taken up by the EVs as opposed to being attached or free floating.
- Figure 3A shows that most buffers are allowing the recovery of 120%+ of EVs. This implies that they are generating more EVs through this process, which should not be happening. Could the authors please reconcile their data collection and analysis, and explain how they got to these numbers?
- If there are high levels of doxorubicin within the EVs, this can reduce the fluorescence read out due to self-quenching. Have the authors considered this within their experiments, and if not, how will they take this into consideration?
- Results section 3.1 include important information for EV isolation and quality controls. This data is in the supplementaries when this should instead be included as a figure on the main manuscript. This will increase the credibility of the authors work and make it less difficult for readers to obtain. Additionally, as figure 1 defines the different parameters and workflow tested in the study, the addition of this information would complement that figure excellently.
- The authors have mentioned that EVs will accumulate specifically in tumour tissue. Could the authors please show primary, non-cancerous cell line experiments where the EVs they are using will not accumulate in these cells?
- Overall, the authors have claimed of an optimised protocol for the loading of EVs, but they have only scratched the surface of these findings by testing in a few cell lines and 1 chemotherapeutic agent. I encourage the authors to test these findings with different known therapeutic molecules like curcumin or paclitaxel. Otherwise, the title should be changed to “Optimised electroporation for loading of extracellular vesicles with doxorubicin”.
Minor concerns and comments:
- Is the level of recovered EVs (Figure 2) normal for standard EV electroporation protocols, is this a main limitation of using electroporation? Can the authors please make it clearer in the text that this is why they’re proceeding with the optimisation (links back to major comment 1).
- Could the authors please clarify the experiment in Figure 2: are these results expected? I would expect that the more EVs loaded would result in more recovered, but nevertheless, could the authors please show how they came to this %?
- Did they use EV depleted media?
- Revise for clarity and conciseness – some expressions and words don’t sound right i.e (Page 2, Line 1 “commend EVs”; “Page 1, “extravagate through leaky vessels”), going so in-depth about WB analysis is not necessary, covering electroporation parameters (Page 2, Line 19) is best suited to methods section and not introduction.
I hope the authors find my comments constructive and helpful in the revision of their manuscript.
Author Response
"Please see the attachment."

Reviewer 2 Report
Dear authors,
I found your study about the optimization of the electroporation technique for EVs loading very interesting, However, there are some aspects that could be improved:
- In the Results section 3.1, I think you could include an image with some of the results about EVs characterization in the main text and not only as supplemental data;
- There several standard deviation and statistical analysis missing in all figures;
- In Figure 2, I cannot see a significant difference between the Control (no electroporation) and the Electroporation (3mM of DOX), so in this case what is the advantage of electroporation? Please clarify.
- In Figure 5, you should also add the control of your EVs loaded with DOX no electroporation.
Author Response
"Please see the attachment."

Round 2
Reviewer 1 Report
Thanks to the authors for their effort effort to address my comments and revise their manuscript, whose overall quality and scientific rigour have been improved. However there are still some of my major concerns that have been adequately dealt with. I therefore would like to bother the author again with these concerns.
Major concerns:
- Study Rationale
- I do believe that the authors should make more effort to readdress my concerns with the study rationale. They have only attempted one of the criteria but really need to expand further. As I mentioned in my previous comments, I strongly suggest the authors to go through (1) Why is electroporation advantageous over other methods (citing other literature is not enough), (2) why it is preferred over other methods, and (3) why it requires optimisation. Without mentioning these information, the introduction is rather disjointed and it is difficult to understand why the authors pursue this study in the first place.
- Additionally, whilst they have begun to explain how pulse programming can affect cargo and EV integrity, and they have mentioned it in the discussion (Page 13), could the authors please expand on this in the introduction to make it clear as to why electroporation needs to be optimised?
- Data interpretation and experimental methodologies
- The authors have mentioned that testing the accumulation of EVs in primary, non-cancerous cell lines would be too difficult. I believe the authors have misinterpreted my suggestions – they can test the uptake of EVs into these cells by similar in vitro assays that they have been doing through their manuscript. There is no need to involve an in vivo model, as increase in vasculature could not be the sole reason behind increased uptake of EVs in target tissues. I do think that it would be important to compare the cytotoxicity of dox-loaded EVs on primary vs tumour cells, which underpins the specificity claim by the authors. Otherwise, the authors will need to revise their angle of cancer-specific treatments if they cannot back up their claims that this is essential.
Minor concerns:
- Clarity and conciseness – extra information that is not necessary and is distracting i.e going so in-depth about WB analysis is not necessary, covering electroporation parameters (Page 2, Line 19) is best suited to methods section and not introduction.
Kind Regards,
Author Response
"Please see the attachment."

Reviewer 2 Report
Dear Authors,
Thank you for your corrections. The overall quality of the paper was improved. However, I must agree with the First Reviewer in the point that should be better discussed the real advantage of electroporation, and what makes this optimization study scientifically sound and useful for the clinical practice.
Best Regards,
Author Response
"Please see the attachment."
